# Treatment of Clival Chordomas: A 20-Year Experience and Systematic Literature Review

**DOI:** 10.3390/cancers15184493

**Published:** 2023-09-09

**Authors:** Carolina Noya, Quintino Giorgio D’Alessandris, Francesco Doglietto, Roberto Pallini, Mario Rigante, Pier Paolo Mattogno, Marco Gessi, Nicola Montano, Claudio Parrilla, Jacopo Galli, Alessandro Olivi, Liverana Lauretti

**Affiliations:** 1School of Medicine, Università Cattolica del Sacro Cuore, 00168 Rome, Italy; carolinanoya1@gmail.com (C.N.); quintinogiorgio.dalessandris@policlinicogemelli.it (Q.G.D.); francesco.doglietto@unicatt.it (F.D.); roberto.pallini@unicatt.it (R.P.); marco.gessi@policlinicogemelli.it (M.G.); nicola.montano@policlinicogemelli.it (N.M.); jacopo.galli@unicatt.it (J.G.); alessandro.olivi@policlinicogemelli.it (A.O.); 2Neurosurgery, Fondazione Policlinico Universitario A. Gemelli IRCCS, Largo Agostino Gemelli, 8, 00168 Rome, Italy; pierpaolo.mattogno@policlinicogemelli.it; 3Otolaryngology, Head and Neck Surgery, Fondazione Policlinico Universitario A. Gemelli IRCCS, 00168 Rome, Italy; mario.rigante@policlinicogemelli.it (M.R.); claudio.parrilla@policlinicogemelli.it (C.P.); 4Pathology, Fondazione Policlinico Universitario A. Gemelli IRCCS, 00168 Rome, Italy

**Keywords:** chordomas, clivus, endoscopic endonasal approach, radiotherapy, surgery, oncology, skull base, survival

## Abstract

**Simple Summary:**

We report the experience of our institution in treating clival chordomas over 20 years and systematically review the recent literature, highlighting factors associated with outcome (age < 50 years, Ki67 ≤ 5%, and adjuvant radiotherapy are associated with better overall survival) and clues for new therapies. Recurrence in clival chordomas remains part of the disease history despite maximal treatment. Still, significant variations are evident in overall and progression-free survival, highlighting the need to develop efficient treatment strategies and recognize which factors reliably predict a more aggressive behavior of clival chordomas.

**Abstract:**

Clival chordomas are rare but aggressive skull base tumors that pose significant treatment challenges and portend dismal prognosis. The aim of this study was to highlight the advantages and limitations of available treatments, to furnish prognostic indicators, and to shed light on novel therapeutic strategies. We conducted a retrospective study of clival chordomas that were surgically treated at our institution from 2003 to 2022; for comparison purposes, we provided a systematic review of published surgical series and, finally, we reviewed the most recent advancements in molecular research. A total of 42 patients underwent 85 surgeries; median follow-up was 15.8 years, overall survival rate was 49.9% at 10 years; meanwhile, progression-free survival was 26.6% at 10 years. A significantly improved survival was observed in younger patients (<50 years), in tumors with Ki67 ≤ 5% and when adjuvant radiotherapy was performed. To conclude, clival chordomas are aggressive tumors in which surgery and radiotherapy play a fundamental role while molecular targeted drugs still have an ancillary position. Recognizing risk factors for recurrence and performing a molecular characterization of more aggressive lesions may be the key to future effective treatment.

## 1. Introduction

Chordomas are rare tumors originating from notochord remnants. In the 2021 WHO classification of Central Nervous System tumors [1], four subtypes of chordoma are listed: conventional, chondroid, dedifferentiated, and poorly differentiated SMARCB1-deficient, the latter affecting primarily pediatric patients. Conventional and chondroid chordomas show a low-grade histopathology and, in the early phases of the disease, have an indolent course and are resistant to conventional photonic radiotherapy and chemotherapy. However, at the later stages, they develop an aggressive clinical behavior with recurrences and metastatic potential [2,3,4,5,6,7]. Dedifferentiated and poorly differentiated chordomas, instead, can have an aggressive course and a high-grade histopathologic appearance since diagnosis. Due to their notochordal origin, chordomas affect the axial skeleton: approximately 50% occur in the sacrococcygeal region, 35% in the skull base and 15% within the mobile spine vertebrae [8]. The estimated incidence is 0.08 per 100,000 per year [5]. In this paper, we analyzed clival chordomas. In that location, tumors, presenting as extradural mass lesions with bony erosion and various grades of dural penetration, often reach sizeable dimensions without producing noticeable symptoms: a VI cranial nerve palsy is the most common clinical sign. The formerly described biological features and bone invasiveness hamper the adoption of a common therapeutic protocol. There are still many open issues about the optimal treatment, mostly related to recurrence management. The natural history of chordomas entails a relatively poor survival of 0.9 years without any treatment [9,10,11]. As first treatment, the gold standard is gross total resection (GTR) with either open or endoscopic surgical approaches, though tumor location and proximity to nervous and vascular structures make GTR challenging, followed by particle therapy [6,12]. Transphenoidal approach is a mainstay of chordoma surgery, allowing wide surgical resection with limited invasiveness [13]; during the last twenty years, a more diffuse use of the endoscopic endonasal approach (EEA) has contributed to the pursuit of GTR with even less morbidity. Instead, the advancements in biomolecular research did only have, so far, a mild impact on clinical outcomes [14,15]. In this paper, we present our institutional surgical series of clival chordomas operated on from 2003 to 2022, with the effort to furnish prognostic indicators; we also provide a systematic review of selected published surgical series as well as an overview of recent advancements in molecular research.

## 2. Materials and Methods

### 2.1. Retrospective Case Series

This retrospective study was conducted on patients operated on for a skull base tumor centered in the clivus in the period between 1 January 2003 and 31 August 2022 at the Neurosurgical Unit of Fondazione Policlinico Universitario Agostino Gemelli IRCCS (Prot. ID 1743). Among those cases, only patients with a pathological diagnosis of chordoma were selected. Demographics, clinical, radiological, and surgical findings, surgical approach, the extent of resection, complications, recurrence, histopathological diagnosis, adjuvant therapy, and relevant outcomes were collected.

Usual clinical management. The following are the basic steps and procedures commonly implemented for all the cases taken into consideration in this paper: (1) preoperative assessment with detailed neuroimaging (MRI and/or CT), neurological exam, and appraisal of the pituitary function if the gland was involved; (2) to obtain the maximal safe tumor resection, we used all the following approaches: microscopic trans-sphenoidal sublabial approach (TSA) in the first 10 years of the analyzed period [13], later on replaced by EEA; open surgical approaches such as pterional, retro-sigmoidal and far lateral approach: the last of these craniotomy approaches was performed in 2013. Intraoperative neuronavigation has been routinely utilized in the last 10 years, as well as intraoperative neuromonitoring, and micro doppler-ultrasound sonography when necessary.

At early postoperative MRI (within 30 days after surgery), the extent of surgical resection was graded as GTR when no residual tumor was evident on postoperative MRI as assessed by a board-certified neuroradiologist, whereas subtotal resection (STR) was defined as a residual tumor of less than 20% of the original mass detected on postoperative imaging. Partial tumor resection (PTR) was defined as a residual tumor greater than 20% of the original mass [2].

At follow-up (3, 6, 12, months and then yearly), any increase in tumor volume was considered as recurrence/progression, and patients were subsequently evaluated by the multidisciplinary board for possible additional treatment. Progression-free survival (PFS) was defined as the time interval between first surgery and first evidence of disease progression or last follow-up. Overall survival (OS) was defined as the time interval between first surgery and death from any cause or last follow-up.

Proton beam irradiation was prescribed after the first surgery or second surgery if not previously performed. Re-irradiation and/or chemotherapy were considered in case of further recurrences or metastasis. CSF leakage was managed with a temporary lumbar drain or with surgical revision.

### 2.2. Literature Review

#### 2.2.1. Surgical Series

A PubMed and Scopus search was performed to identify reports and clinical series of patients with skull base chordoma treated using both endoscopic endonasal resection and microsurgery resection. The PRISMA review method was applied as shown in Figure 1.

A systematic search was performed in December 2022 by using PubMed and Scopus for articles published between January 2000 and October 2022. We searched both databases for all articles stating “chordoma” plus “surgery” plus either “clivus”, “clival”, or “skull base”. The inclusion criteria were monocentric surgical series of clival chordomas including more than 25 adult patients. We limited results to only articles in English, involving adult patients, and published between 2000 and 2022. The document screening is schematically illustrated in Figure 1.

After duplicates were removed, all identified articles were independently assessed for screening by two reviewers (CN, LL) based on their titles and abstracts. Articles related to surgical technique for chordomas or dealing generically on skull base tumors including chordomas, or multi-institutional series, were not included. Similarly, articles related to chordomas of mixed anatomical locations were not utilized, as well as pediatric series. As a next step of the research, based on a complete review of each article, we excluded those the results of which were outside the scope of our research. These included results mixing chordomas and chondrosarcoma and/or other skull base pathologies, or not reporting the amount of tumor removal. Papers that did not report the surgical technique utilized (endoscopy, type of craniotomy) were also excluded.

We then gathered the outcomes of these surgeries including the rate of GTR, mean follow-up, rate of recurrence, and outcome. Finally, we extracted information regarding the most common complications of chordoma surgery: cranial nerve injury, CSF leak, and meningitis.

#### 2.2.2. Molecular Studies

This search was performed on the “Web of Science” (WOS) database to identify original molecular studies published in the period of 2010–2022 with ≥30 citations. We decided to use the WOS database which adopts more stringent criteria for reporting citations. The PRISMA review method was utilized as shown in Figure 2 and search terms were “molecular and chordomas”. Reviews were excluded, as well as case reports. The analysis was then conducted with the same modalities described previously.

## 3. Results

### 3.1. Review of the Literature

#### 3.1.1. Surgical Series

Major findings are summarized in Table 1.

The systematic search resulted in 469 articles, of which 292 were records from PubMed and 177 were records from Scopus. After title and abstract screening, 286 articles were excluded, resulting in 183 articles for full-text evaluation. The full-text evaluation excluded 11 reports regarding pediatric cases, 2 multicentric studies, 39 case reports, 29 case series that enrolled less than 25 patients, 20 reports analyzing several pathologies affecting the skull base, 14 technical notes or technical videos introducing surgical techniques. In total, 56 reports were excluded and categorized as “other reasons”; of those, common reasons for exclusion during screening included review and meta-analysis articles, spinal cases mixed to skull base pathology, and case series limited to medical and radiotherapy treatment. After excluding two papers because the full text was not available, finally, ten articles were included in the pooled analysis (Figure 1, Table 1).

#### 3.1.2. Molecular Studies

Major findings are summarized in Table 2.

The availability of cellular lines derived from sacral (UCH1) [3] and, more recently, from clival chordomas (UM-Chor1) [4] surgeries has been of paramount importance for investigating genetic and epigenetic oncogenic mechanisms, and for identifying the molecular features that can be the object of targeted therapies. This can be obtained by utilizing cells resulting from the patients’ tumoral tissue collected during surgery. Telomerase-positive aggressive chordomas are ideal candidate for cell line generation [37].

### 3.2. Institutional Series

Between January 2003 and December 2022, 42 patients (22 males and 20 females; median age 50.8 ± 18.4 years, range 17–85 years) with clival chordoma were treated with 85 surgeries (microsurgical open and/or endoscopic endonasal procedures). Three patients were excluded from statistical analysis because information of their follow-up was not available.

Twenty patients (47.6%) underwent a single procedure, while twenty-two (52.4%) patients had multiple surgical treatments, ranging between 2 (21.4%) and 7 procedures (2.4%). Two-step surgery was chosen for three patients. Seven patients had their first surgery in another hospital and were referred to us for the treatment of a recurrence. Most lesions were located in the superior and middle clivus.

Clinical findings are detailed in Table 3. The most common symptom at presentation was diplopia (23 patients, 51.2%). Symptoms and signs of visual deficit were observed in three patients (7.2%), while four patients (7.5%) discovered the tumor incidentally and had no related symptoms. Of the 42 patients, 7 underwent craniotomy approach (16.7%); 19 patients (45.2%) had TSA and 16 patients (38.1%) had EEA. Histological findings are reported in the table above (Table 3). Among the 22 patients who underwent multiple surgeries, the proliferative index, evaluated by Ki-67 staining, increased after each surgery in 3 patients. All but eight patients who had undergone primary treatment were treated with postoperative proton therapy or other adjuvant therapy; three patients had adjuvant chemotherapy including off-label treatments (bevacizumab, sirolimus).

The median follow-up in our series was 15.8 years. Survival rates were 70.6% at 5 years, 49.9% at 10 years and 16.1% at 20 years. PFS was 37.2% at 5 years and 26.6% at 10 years.

Kaplan–Meier analysis of prognostic factors for OS and PFS is shown in Figure 3, Figure 4, Figure 5, Figure 6, Figure 7 and Figure 8.

A significantly improved OS was observed in younger patients (median OS of 18 years in patients younger than 50 years vs. 5.4 years in those older than 50 years, *p* = 0.0054; Figure 3), in tumors with Ki67 ≤ 5% (median OS of 18 years vs. 4 years in those with Ki67 > 5%, *p* = 0.0148; Figure 4), and when adjuvant radiotherapy was performed (median OS of 18 years vs. 3.3 years without radiotherapy, *p* = 0.0001; Figure 5). Conversely, the surgical approach was not correlated with survival, though median OS was non-significantly inferior in patients operated on with craniotomy as first approach (4 years) compared with those operated on using EEA or TSA (15 and 9.4 years, respectively; Figure 6). Multivariate Cox analysis showed an independent prognostic role for the OS of postoperative radiotherapy (*p* = 0.0248; Table 4).

Significant prognosticators of PFS were n of surgeries (median PFS of 5 years if up to two surgeries vs. 2 years if >2 surgeries, *p* = 0.0084; Figure 7), and proliferative index (median PFS 2 years if >5% and 5 years if ≤5%; Figure 4). The timing of adjuvant radiotherapy, i.e., early or delayed, did not impact the OS; instead, conceivably, patients who had early adjuvant radiotherapy had an improved PFS than those undergoing delayed radiotherapy (Figure 8).

By comparing patients reaching a 10-year OS vs. those not reaching a 10-year OS, we confirmed the positive prognostic value of age, proliferative index and radiotherapy.

In detail, (*i*) age was >50 years in 18.8% patients surviving at least 10 years vs. 73.7% patients surviving less than 10 years (*p* = 0.0020, Fisher exact test); (*ii*) proliferative index was >5% in 0% cases reaching a 10-year OS vs. 45.5% of those cases not reaching a 10-year OS (*p* = 0.0379, Fisher exact test); (*iii*) radiotherapy was performed in 92.3% of cases surviving at least 10 years vs. 50% of cases surviving less than 10 years of OS (*p* = 0.0329, Fisher exact test). Instead, the type of surgery and the number of operations did not impact the 10-year OS. Multivariate logistic regression analysis showed an independent prognostic role of age in determining the 10-year OS (*p* = 0.0065).

## 4. Discussion

Despite their low incidence (<1/100.000 [5]) and the fact of being slow-growing tumors, chordomas deserve attention because of their aggressive behavior: they are substantially resistant to conventional chemo-radiotherapy, and recurrences have been reported as high as >50% at 10 year after first surgery [6]. GTR and heavy-particle radiotherapy affect PFS but, notwithstanding these treatments, many cases do recur [6,8]. In the literature, there is an abundance of studies aiming at identifying molecular prognostic indicators and furnishing specific targets for new therapies, some of which come from our research group [9,14,24,38]. These studies have deepened our knowledge of chordoma features, as evidenced by our literature review (Table 2); however, the clinical relevance of those findings is still low [15]. For instance, RTKs and EGFR expression prompted the use of combined therapies such as Imatinib plus Sirolimus that have shown only weak results in selected, advanced cases of multiple recurrences.

Considering also the contribution of less cited recent biomolecular studies, it would seem clear that (1) cell lines obtained from sacral and clival chordomas constitute a fundamental and useful tool for researchers to obtain further biomolecular information; (2) conventional and poorly differentiated chordomas express brachyury while dedifferentiated types do not, and this characteristic limits the potential of an antibrachyury vaccine [10,11]; (3) identification of epigenetic changes such as miR hyper-hypo-expression in recurrent tumors might furnish new therapeutic options [3,9]. Genetic and gene expression changes identified using NGS could provide further potential therapeutic targets [39]. Overall, given the lack of proven evidence, and since CH surgery is performed in referral centers, it is important to promote collaborative molecular studies on patient tumoral tissue that could identify targets for tailored therapies to treat recurrences.

There are some issues that have been widely discussed in the literature on clival chordomas. In brief, the patient’s age, tumor location (upper, middle or lower clivus, parasellar extension, or involvement of multiple skull base regions), tumor volume, extent of surgical resection, proliferation index, and adjuvant heavy-particle radiotherapy have been recognized influencing PFS and OS [6,8,12], even if distinctions among authors have been reported.

What we found lacking in the literature is an estimation of the risk period for recurrence and, consequently, a prevision of when that hazard ceases and when, eventually, patients might be considered as cured.

Then, our aim was mostly to investigate which factors were associated with prolonged PFS and OS, and which, if any, to cure.

### 4.1. Not Controversial Issues

#### 4.1.1. Age

The average age of patients in the present study on the first appearance of symptoms was 50.1 years, which is comparable with the figure reported by Passeri et al. (50 years), and it is lower than the average age reported by Crockard et al. (58.1 years) [10,16].

Consistent with what has been reported in other surgical series, age was found to be an independent prognostic indicator with a cut-off of 50 years old in our series. Moreover, that age limit is emphasized when analyzing 10-year OS: in fact, 70% of patients with OS < 10 years were >50 years old. Then, younger patients have a higher chance of OS > 10 years. Though the prognostic role of age is well established, it must be disclosed that we did not consider chordoma-specific mortality.

#### 4.1.2. Pathology (Proliferative Index)

Proliferative index, as assessed using Ki-67 or MIB-1 staining, has been recognized as strictly related to prognosis in CH patients, influencing both PFS and OS in our series (Figure 4). Interestingly, proliferative index had a particular relevance in the group of long survivors: in particular, all patients with OS > 10 years had tumors with a proliferative index of less than 5%. The literature data widely confirm the prognostic role of proliferative index, though other molecular players influence prognosis as well [26,40]. The failure of proliferative index in retaining an independent prognostic role at multivariate analysis in our series could reflect the role of such other markers [40], which have not been studied in detail in the present work, or the overwhelming impact of radiotherapy.

#### 4.1.3. Radiotherapy

The role of radiotherapy in prolonging PFS and OS in chordoma patients is widely recognized. Chordomas are considered relatively resistant to conventional radiotherapy. In order to overcome chordoma radioresistance and to obtain satisfactory local control, stereotactic radiotherapy or heavy particles (hadrons) have been employed. Hadrons are high-dose protons or charged particles such as carbon ions, helium or neon. Hadron-based therapy allows the delivery of higher doses of radiation to the tumor with minimal injury to the surrounding tissues and improved radiobiological effect [41,42]. Therefore, hadron therapy offers a potential survival advantage compared with traditional photon therapy, including improved effectiveness and reduced delayed adverse events. The high-dose volume should include any macroscopic disease as well as surgical margins, while the low-dose volume should encompass areas at risk of microscopic spread, skip metastases, or seeding due to surgical procedures. The main clinical complications due to radiation therapy are those involving the visual apparatus and pituitary insufficiency [42].

Aside from a few reports describing the similarity of results obtained comparing conventional radiotherapy and proton-beam therapy, it would seem that there is a superiority of treatments with heavy particles. In our series, the role of proton-beam radiotherapy was unequivocal, having a positive influence on survival independently from the timing of its administration (i.e., after the first or second surgery).

### 4.2. Controversial Issues

#### 4.2.1. Surgery

Various surgical approaches have been described for the resection of clival chordomas, including anterior transcervical retropharyngeal, transseptal–transsphenoidal, pterional, retromastoid, lateral suboccipital, subfrontal, extended frontal, transbasal, subtemporal–infratemporal, presigmoid–subtemporal, transpetrosal, lateral transcondylar [43] and transoral–transpalatopharyngeal [44]. Since the 1960s–70s, the transsphenoidal approach has been proposed for these tumors [13], with the natural nasal corridor advocated as the most direct route to the clivus. The development of endoscopic endonasal technique, starting from the pioneering report by Jankowski et al. [45] in 1992, and the development of extended endoscopic approaches further fueled the use of the transsphenoidal route in the treatment of clival chordomas. Indeed, extended EEAs demonstrate equivalent or superior resection rates with respect to other surgical routes, with significantly reduced invasiveness, including fewer sequelae, complications, and mortality [8,24,46,47,48]. Decision-making regarding surgical approach and the extent of resection must be undertaken as a cost–benefit analysis with consideration of a multitude of patient- and tumor-specific factors, including tumor location, the neurovascular anatomy involved and patient functional status, among others, with the goal to achieve as radical a resection as possible at first presentation while avoiding morbidity [16,17,23].

Our systematic review of the literature showed a 10% mean increase in the rate of GTR of EEA compared to that of open surgery (Table 1) [17,22,23,24]. This overall difference in GTR is what likely led to a vast difference in the rate of recurrence, with endoscopic surgery resulting in over half the mean rate of recurrence across assessed studies [8].

In our series, EEA, widely used in the last ten years, allowed GTR for most clival chordomas during one-step surgery, reducing the hospitalization time and the necessity of further surgeries. However, overall, among our patients, the type of surgical approach did not influence PFS and OS in the subgroup of patients with OS > 10 years.

Remaining in the field of surgery, contrary to what has been reported in the literature, in our series, the statistical evaluation revealed that surgical complications did not affect OS [24]. This finding may be explained by the low incidence of major complications that we observed. In fact, in our series, surgical complications with fatal outcomes (rupture of main arteries and meningitis from multi-resistant germs) occurred in four patients, and in three of them during treatment of re-recurrences.

Then, it is important to reiterate that surgery for recurrent chordomas is a high-risk procedure; however, experienced surgeons and the use of up-to-date instrumentation might reduce the incidence of fatal complications.

#### 4.2.2. Recurrences and Definition of Cure

The long median follow-up of our series prompted us to reflect not only on the percentage of recurrences that is normally reported in the literature [8,16,21,22]: instead, our effort was aimed to analyze the timing of recurrences to figure out how long the hazard of cancer regrowth persists, and if and when we could consider a patient as cured. These aspects are of paramount importance; in fact, having referral periods for “risk of recurrence” or “out of risk of recurrence” would help in stratifying patients during follow-up and deciding the timing of radiological exams.

In our series, when present, the first recurrence always occurred 10 years after surgery. Therefore, it is realistic to speculate that a PFS (defined here as GTR without recurrences) of 11 years from first surgery is predictive of cure.

On the other hand, we evidenced that the rate of multiple recurrences was still compatible with survival of as long as 15 years from the first recurrence in our series. We explained that observation as the expression of a good clinical conduct, with lower surgical morbidity and an overall better management of complications so OS was not affected. It is evident that these results can be gained in referral centers, which have experience in dealing with such rare tumors.

### 4.3. Limitations of the Study

We did not consider in detail other factors influencing the recurrence of clival chordomas such as tumor volume, tumor doubling time, and tumor location; those aspects have already been evaluated in the literature, and therefore, we decided to focus our attention on a few more specific aspects. However, we are aware that the above choice might weaken our findings.

## 5. Conclusions

There are still few bio-molecular features strictly predictive of the outcome of clival chordomas and few therapeutic tools for aggressive forms.

However, the possibility of radically treating chordomas by implementing a combination of surgery and heavy-particle radiotherapy is real, as shown by the elements we provided as a useful tool for clinicians.

The individuation of “high-risk period vs. low-risk period for recurrences” would also help in planning a shrewd follow-up.

## Figures and Tables

**Figure 1 cancers-15-04493-f001:**
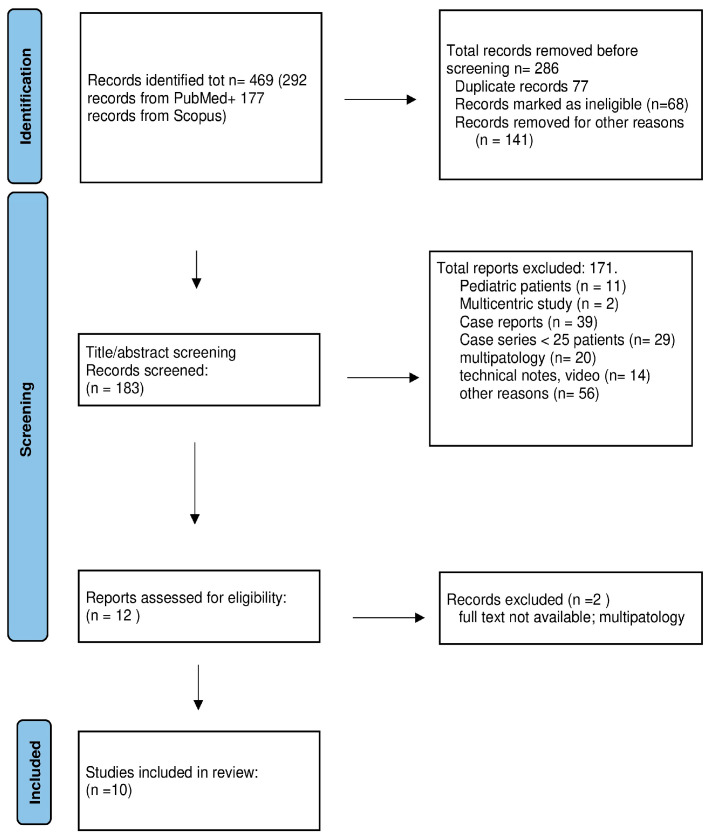
PRISMA flow chart of surgical series published in the period of 2000–2022.

**Figure 2 cancers-15-04493-f002:**
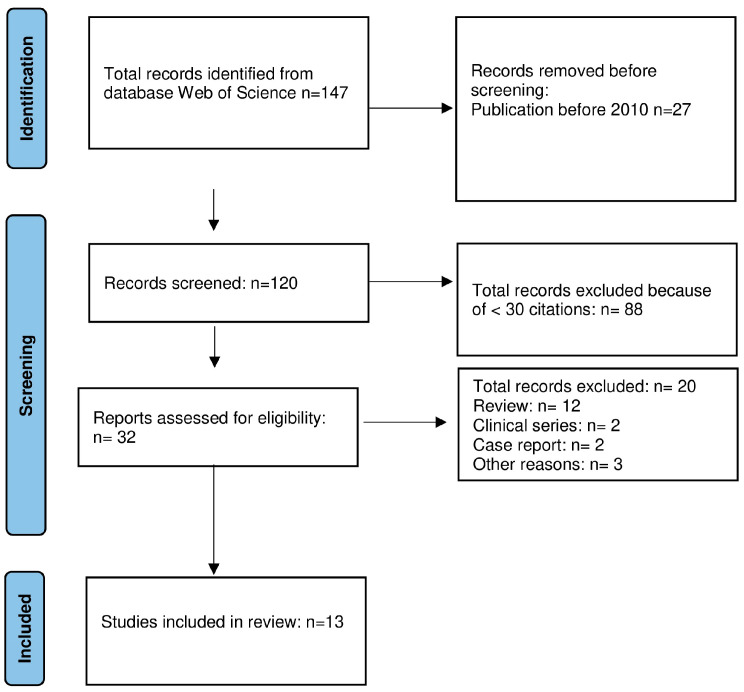
PRISMA flow chart of molecular studies published in the period of 2010–2022.

**Figure 3 cancers-15-04493-f003:**
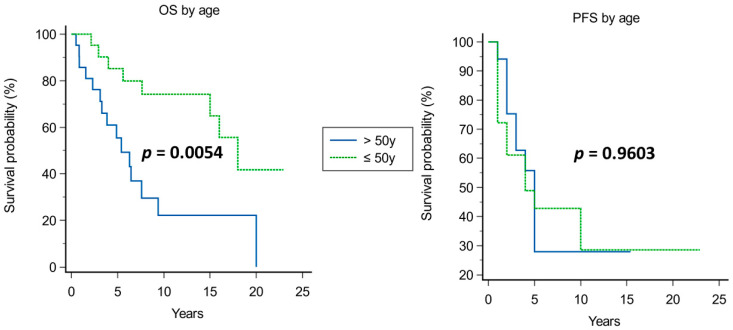
Kaplan–Meier curves for of OS (**left**) and PFS (**right**) based on age.

**Figure 4 cancers-15-04493-f004:**
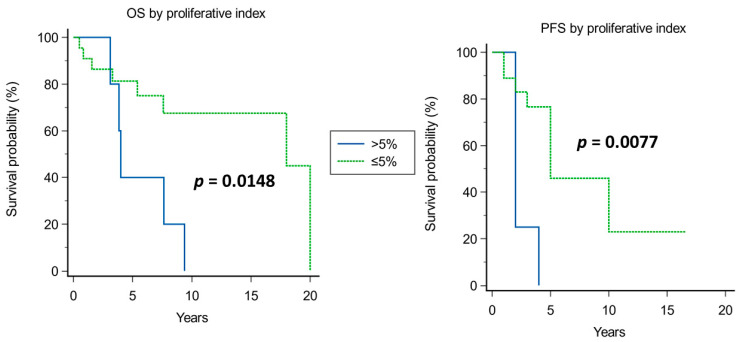
Kaplan–Meier curves for OS (**left**) and PFS (**right**) based on proliferative index (cut-off > 5%).

**Figure 5 cancers-15-04493-f005:**
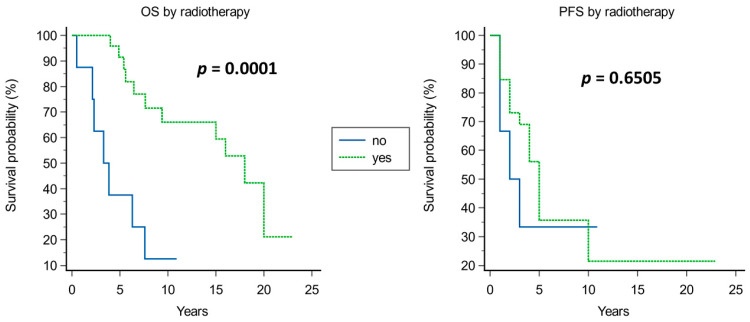
Kaplan–Meier curves for OS (**left**) and PFS (**right**) based on adjuvant radiotherapy.

**Figure 6 cancers-15-04493-f006:**
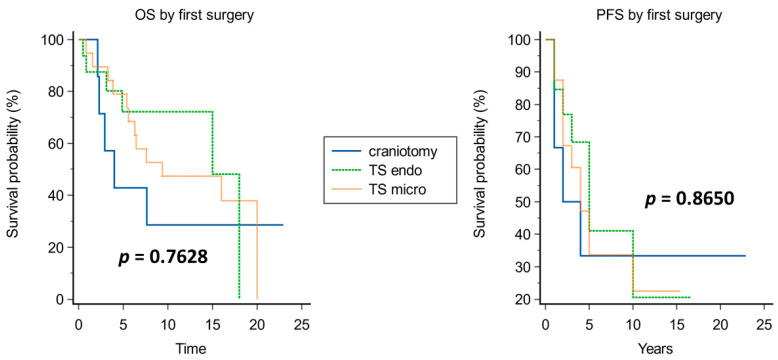
Kaplan–Meier curves for OS (**left**) and PFS (**right**) based on approach at first surgery.

**Figure 7 cancers-15-04493-f007:**
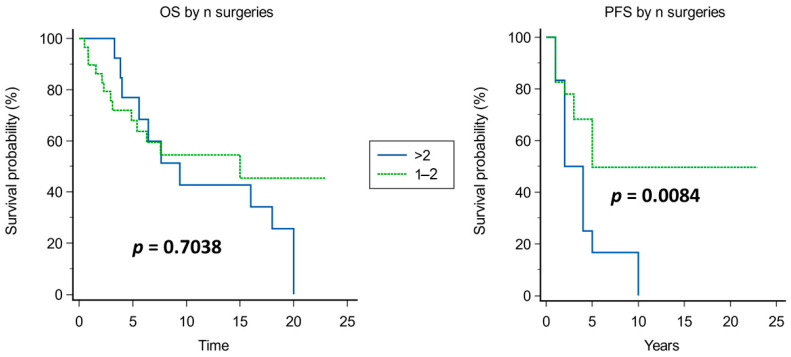
Kaplan–Meier curves for of OS (**left**) and PFS (**right**) based on n of neurosurgical operations.

**Figure 8 cancers-15-04493-f008:**
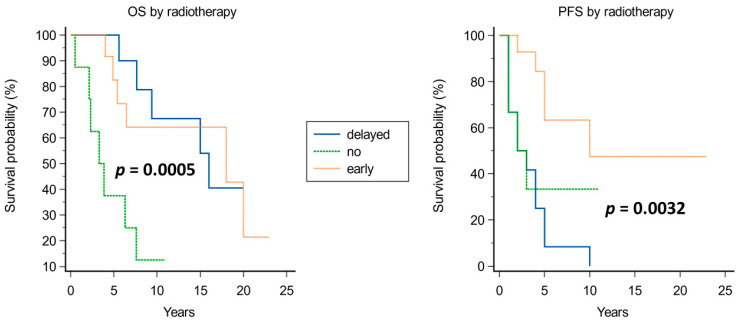
Kaplan–Meier curves for OS (**left**) and PFS (**right**) based on radiotherapy timing.

**Table 1 cancers-15-04493-t001:** Systematic review of surgical series.

Author, Year	N. of Patients	Surgical Approach	Complications	GTR	Follow-Up (Months)	Recurrence	Outcome
Samii, 2007 [16]	49	Open (49)	CSF leak 5.4%, meningitis 5.4%	49%	63	NA	5-year OS 65%
Koutourousiou, 2012 [17]	60	Endoscopic (NA), open (NA)	CSF leak 20%, CN injury 6.7%	67%	17.8	20%	NA
Ouyang, 2014 [18]	77	Endoscopic (NA), open (NA)	Overall 27.8%, CN injury 18.2%	33%	60	NA	3-year PFS 92.0%
Jahangiri, 2015 [19]	50	Endoscopic (34), open (9), combined (7)	CSF leak 12%, CN injury 6%, meningitis 12%	52%	41	51%	NA
Zhang, 2016 [20]	32	Endoscopic (32)	CSF leak 12%	28%	20	NA	5-year PFS 16.5% 5-year OS 69.5%
Raza, 2018 [21]	29	NA	NA	41%	28	NA	Disease-specific survival 44.4 months
Zoli, 2018 [22]	65	Endoscopic (65)	CSF leak 2.5%	59%	52	40%	NA
Wang, 2020 [23]	49	Endoscopic (49)	CSF leak 30.1%, CN injury 5.5%	64–85%	41.5	14%	Mortality 12%
Bai, 2022 [8]	284	Endoscopic (349), open (31)	CSF leak 3.9%	40%	43.9	55%	5-year disease-specific survival 71.0%
Passeri, 2022 [24]	210	Endoscopic (142), open (123)	CSF leak 12.1%; CN injury 17.7%	44%	59.2	42%	5-year PFS 52.1% 5-year OS 75.1%

CN, cranial nerve; CSF, cerebrospinal fluid; GTR, gross total resection; NA, not available; OS, overall survival; PFS, progression-free survival.

**Table 2 cancers-15-04493-t002:** Systematic review of molecular studies, major findings.

Author, Year	Material		Main Findings
	Bank Cells	Patients’ Tissues	
Tamborini et al., 2010 [25]		Sacrum, spine, and clivus chordomas	The existence of an autocrine/paracrine loop involving some imatinib-related RTKs
Horbinski et al., 2010 [26]		Skull base chordomas	Identification of biomarkers with a prognostic role
Le et al., 2011 [27]		Spine and skull base chordomas	High genomic instability (large copy number losses)
Shalaby et al., 2011 [28]	U-CH1	Sacrum, spine, and skull base chordomas	The importance of molecular studies for targeted therapies (like EGFR antagonists)
Aydemir et al., 2012 [3]	U-CH1	Chordomas, nucleus pulposum	Characterization of chordoma stem cells
Bayrak et al., 2013 [29]	U-CH1	Chordomas, nucleus pulposum	Identification of down and upregulated miRNAs
Kitamura et al., 2013 [30]		Skull base chordomas	Identification of specific genetic/molecular and clinical prognostic factors
Choy et al., 2014 [31]		Sacrum, spine, and skull base chordomas	Point mutations in tumor suppressor genes
Scheil-Bertram et al., 2014 [32]	U-CH1, 2	Chordomas, nucleus pulposum	Identification and validation of genes involved in chordomas genesis
Zhang et al., 2014 [33]	U-CH1, U-CH2	Clival chordomas	Identification of down-regulated miRNAs as a potential therapeutic tool
von Witzleben et al., 2015 [34]	U-CH1, 2, 3, 6, 7, 11, 12	Sacrum	Development of new cell lines and evaluation of CDK4/6 inhibitors
Wang et al., 2016 [35]		Sacrum and spine chordomas	Alterations of chromatin regulatory genes (SETD2)
Bai et al., 2020 [36]		Clival chordomas	Role of LncRNA in dural penetration

CDK, cyclin dependent kinase; EGFR, epidermal growth factor receptor; RTK, Receptor Tyrosine Kinase; SETD2, SET domain containing 2.

**Table 3 cancers-15-04493-t003:** Clinical findings of patients suffering from clival chordoma treated between January 2003 and December 2022.

Pt N#	Age	Sex	Symptom at Onset	N° of Surgeries	Surgical Approach	EOR	Complications	Adj. RxTp after First Surgery	Histology	Proliferative Index at First Surgery	Recurrence	Treatment of Recurrence	Pt Outcome (Yrs)
1	55	F	diplopia	1	EEA	GTR	-	yes	chordoma pan-CK+/S100+/vimentin+	1–2%	no		AWD (4)
2	23	M	diplopia	1	EEA	GTR	-	yes	chordoma s100+/EMA+/hTERT+/p53-	1.5%	no		AFD (17)
3	57	F	incidental	1	EEA	GTR	-	yes	chordoma pan-CK+/S100+/vimentin+/EMA+	5%	no		AWD (3)
4	29	M	diplopia	1	EEA	GTR	-	yes	chordoma pan-CK+/S100+/vimentin+/EMA+	1%	no		AWD (3)
5	32	M	incidental	1	EEA	STR	-	NA	chordoma pan-CK+/S100+/vimentin+	2%	NA		NA
6	41	M	diplopia, loss of vision	1	EEA	GTR	-	yes	chordoma S100+/vimentin+/brachyury+	NA	no		AFD (10)
7	80	F	diplopia	2	EEA; STA	STR	-	no	chordoma pan-CK+/S100+/vimentin+/EMA+	5%	Yes @ 3 yrs	EEA + RT	AWD (6)
8	59	M	nystagmus, diplopia	1	EEA	GTR	CSF leak	no	chordoma pan-CK+/S100+/vimentin+	3%	no		DD (1)
9	40	M	diplopia, headache	1	EEA	STR	-	yes	chordoma pan-CK+/S100+/vimentin+	3%	yes @ 5 yrs	RT	AWD (7)
10	53	M	diplopia	1	EEA	GTR	-	yes	chordoma pan-CK+/vimentin+	NA	no		DOD (5)
11	76	F	dysphonia	1	CR	GTR	-	yes	-	<2%	no		DOD (6)
12	54	F	diplopia	4	EEA	PR	hearing loss; bilateral trigeminal pain	yes	chordoma pan-CK+/vimentin+	NA	yes @ 4 yrs	STA + RT+ sirolimus (7 mL/die) and erlotinib (150 mg/die); @3 yrs STA; EEA	DD (7)
13	43	F	diplopia	5	EEA	STR	ophthalmoplegia, hypovisus	no, not indicated due to rapid residual growth	chordoma pan-CK+/S100+/vimentin+	2%	Yes @ <1 yrs	EEA + RT; EEA + RT; NA (other hospital), NA (other hospital)	AWD (6)
14	85	M	incidental	2	EEA	STR	-	NA	chordoma pan-CK+/S100+/vimentin+	5–10%	yes @ 2 yrs	EEA	DOD (3)
15	23	M	diplopia	3	EEA	PR	-	yes, proton + Imatinib	chordoma pan-CK+/S100+/vimentin+	2%	Yes @<1 yrs	EEA + C0 − C4 fixation, sirolimus (7 mL/die) and erlotinib (150 mg/die); CR	DOD (18)
16	69	M	diplopia	2	EEA	GTR	-	no	chordoma pan-CK+/vimentin+/S100+	2–3%	yes @5 yrs	EEA + RT	AWD (7)
17	23	M	incidental	1	CR	GTR	-	yes	chordoma CAM 5.2+/Vimentin+/S100+	NA	NA		AFD (17)
18	82	F	diplopia	2	STA	STR	visual impairment bilaterally, right VI CN palsy, ipsilateral hearing loss	no	chordoma	NA	yes @1 yr	STA	DD (2)
19	41	M	diplopia	1	CR	GTR	-	yes	chordoma	NA	no		AFD (24)
20	42	M	V2 hypoaesthesia, dysphagia	1	CR	GTR	right VI CN palsy; obstructive hydrocephalus requiring CSF shunting	no	chordoma pan-CK+/S100+/vimentin+	NA	no		DD (2)
21	40	M	diplopia	3	CR	STR	left VII and VIII CN palsy and dysphagia; hemiplegia	yes + Cyber knife	chordoma pan-CK+/S100+/vimentin+/p53+ 30%	8–10%	yes @ 2 yrs	CR; @<1 yr CR	DD (4)
22	43	M	left tinnitus	1	STA	GTR	CSF leak	NA	chordoma pan-CK+/S100+/ p53 +/brachyury+	3–4%	no	-	AFD (11)
23	60	F	rhinosinusitis	1	STA	GTR	CSF leak	NA	chordoma pan-CK+/S100+/vimentin+	NA	no		AFD (16)
24	39	F	diplopia	4	STA	GTR	-	no	chordoma	NA	yes @4 yrs	CR; @5 yrs STA+ RT; @3 yrs CR	DOD (30)
25	72	M	incidental	1	STA	GTR	seizure	no	chordoma pan-CK+/vimentin+	2%	no		DOD (7)
26	68	F	diplopia	1	STA	GTR	-	NA	chordoma pan-CK+/S100+/vimentin+/p53 15%	1%	no		DOD (2)
27	31	M	diplopia	4	STA	GTR	obstructive hydrocephalus requiring CSF shunting; hemorrhagic infarct	no	chordoma p53+	NA	yes @ 2 yrs	STA+RT; @4 yrs EEA+ Cyber Knife; @2 yrs CR	AFD (20)
28	51	M	V2 hypoaesthesia	5	STA	GTR	hypopituitarism, meningitis	no	chordoma pan-CK+/S100+/vimentin+	3%	yes @ 5 yrs	CR+ RT; @ 12 y STA + RT; @1 yrs EEA; @1 yr EEA + Sirolimus	DD (20)
29	60	M	diplopia	3	STA	STR	hypovisus	no	chordoma pan-CK+/S100+/vimentin+	2%	yes @ 2 yrs	EEA; @1 yr EEA	DOD (3)
30	71	F	diplopia	2	STA	STR	NA	no	chordoma	NA	yes @ 3 yrs	STA	DD (4)
31	48	F	diplopia	2	STA	STR	-	no	chordoma pan-CK+/vimentin+	4–5%	yes @ <1 yr	STA + RT	AFD (11)
32	66	F	diplopia	2	STA	GTR	-	no	chordoma pan-CK+/S100+/vimentin+/h-TERT+/p53+ 6–8%	2%	yes @ 5 yrs	STA	AFD (17)
33	55	F	diplopia	4	STA	STR	bilateral VI CN palsy	yes	chordoma pan-CK+/S100+; p53+; hTERT+;	1–12%	yes @ 2 yrs	STA; @ 2 yrs STA + RT; STA + chemo	DD (10)
34	47	F	headache	3	STA	GTR	CSF leakage; hypopituitarism	no	chordoma S100+/vimentin+	1%	yes @ 10 yrs	STA; @ 9 yrs EEA + RT	AWD (20)
35	33	M	headache	1	CR + c0-c2 fixation	NA		NA	chordoma pan-CK+/vimentin+/S100+	NA	NA		DD (3)
36	26	F	tinnitus, left tongue fasciculation	7	CR + c0-c2 fixation	STR	VII to XII CN palsy	RT+oxygen-ozone therapy and Temodal and homeopathic anticancer therapies	chordoma pan-CK+/S100+/vimentin+; p53+	5–6%	yes @ 4 yrs	CR; (second surgical time) CR; CR + Sirolimus; EEA and transoral combined; @1 yr CR; EEA; STA + Radiotherapy	DD (8)
37	44	M	V3 hypoaesthesia, ptosis	5	STA	STR	hemiparesis, VI CN palsy; obstructive hydrocephalus requiring CSF shunting	no	chondroid chordoma; VEGF+, EGFRvIII+	NA	yes @ <1 yr	STA; Cyber knife; STA; EEA; EEA + Bevacizumab + Erlotinib	DD (5)
38	37	M	diplopia	2	EEA	STR	-	no	chordoma; HTERT+; p53+	NA	yes @ 1 yr	EEA	DD (13)
39	79	F	hypovisus	6	STA	NA	-	no	chordoma ck+/S100+/p53+; vimentin+/hTERT-;	4–8%	NA		DOD (3)
40	66	F	NA	2	STA	NA	-	NA	chordoma pan-CK+/S100+/vimentin+	3%	NA		NA (1)
41	65	F	NA	1	EEA	NA	-	NA	chordoma pan-CK+/S100+/vimentin+	NA	NA		NA (1)
42	36	F	headache, diplopia	1	STA	NA	-	NA	chordoma pan-CK+/S100+/vimentin+/p53-	3%	NA		AFD (20)

AFD, alive free from disease; AWD, alive with disease; CN, cranial nerve; CSF, cerebrospinal fluid; CR, craniotomy; DD, died of disease; DOD, died of other diseases; EEA, Endoscopic endonasal approach; GTR, gross total resection; NA, not available/applicable; PR, partial resection; Pt, patient; RxTp, radiotherapy; STA, sublabial transphenoidal approach; STR, subtotal resection; Yrs, years.

**Table 4 cancers-15-04493-t004:** Cox multivariate analysis of prognostic factors for OS.

Covariate	B	SE	Wald	*p*	Hazard Ratio	95% CI
Age	−0.9215	0.4708	3.8307	0.0503	0.3979	0.1581 to 1.0013
Proliferative index	0.9323	0.5355	3.0306	0.0817	2.5402	0.8893 to 7.2562
Radiotherapy	1.1323	0.5045	5.0372	0.0248	3.1028	1.1543 to 8.3409

## Data Availability

All the material is owned by the authors and no permissions are required.

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
