# Peer review of "Treatment of Clival Chordomas: A 20-Year Experience and Systematic Literature Review"

_cancers, 2023, doi:10.3390/cancers15184493_

Round 1

Reviewer 1 Report

Skull base cancers review

In this manuscript the authors review their institution experience with clival chordoma and review the literatures. This is an important set of data to support modern approaches to skull base chordoma.  This is of high interest to the field as 1. The chordoma community is working to standardize therapeutic approaches (GTR plus/minus adjuvant RT) and 2. We need more data on the natural history of this disease to support drug development.  The paper needs a bit of work but will be publishable with edits. I look forward to reading the next version. 

1.     English editing for word flow needed throughout

2.     In the intro paragraph it is important to distinguish the subtypes upfront.  Dediff and poorly differentiated chordoma are not typically indolent (ever). 

3.     P2 line 49: please include poorly differentiated chordoma, now recognized by the WHO as a subtype. This subtype has loss of INI1, which you may want to note. It is also more common in children and young adults

4.     P2 line 59 – would delete heavy particles as I believe ‘particle therapy’ is the modern term

5.     Consider removing the abbreviation CH (chordoma). That isn’t a standard abbreviation outside of cell lines

6.     P2 line 73 – just conventional chordoma? Other subtypes? The other subtypes should be analyzed separately

7.     Page 4 L 120, would swap out useless for another word. How about “we excluded those whose results were outside the scope of our research” 

8.     Table 2: you are missing some sequencing papers - PMID: 28860410 table 2; PMID: 28860410

9.     Table 2: does this add anything to your paper? I would consider deleting all sequencing and cell line references. This is a clinical paper and it isn’t really needed

10.  Table 3 has a case change in some areas

11.  P13L184 accidentally should be incidentally. 

12.  P13L192 – was this all-cause mortality or just chordoma-related mortality? Please specify. If all-cause, then the OS by age is misleading.  What happens when you divide into quartiles? It may be just that the people over 75 are driving this observation, not the cancer.  For example, the PFS is the same between the two groups suggesting this is not related to chordoma 

13.  The proliferation index findings are notable.  Please make sure to include reference to the UPMC data in this space. Can you comment on the proliferation index P score when looking at multivariate analysis? I’m surprised that isn’t significant

14.  Can you explain your hypothesis on Figure 5?  Why would there be no difference in PFS but such a marked difference in OS? 

15.  P15L230 this 10-year OS v non- 10year OS is very confusing. Please reword.

16.  Page 15 line 242 GTR and heavy particles…do not protect from recurrences. I think that is dated literature.  Your data, and other modern series, would argue that this reduces recurrence. In fact, you state this on line 288

17.  Discussion: I would rework the discussion.  The 2nd paragraph on cells lines, vaccines, miR, is outside the scope of this paper. I would focus on the surgical and RT part, plus your identified risk factors.

18.  Line 283 this should be less than 5% correct?

19.  How do your results change over time? i.e is modern pfs better than >10 years ago, suggesting current surgical approaches are better?

20.  Is there any indication that low proliferation rate index pts can avoid radiation?

Needs an editor for some style/word flow, etc

Reviewer 2 Report

I congratulate authors for sharing their experience with clival chordomas and preparing the systematic review. 

The retrospective review is adequate including surgical treatments before and after endonasal endoscopic era. Interestingly, endoscopic endonasal approaches were not correlated with increased survival or as a favorable prognostic factor. 

The molecular biology segment is very welcomed and providing additional value to the retrospective review. 

I recommend expanding the discussion on the different radiation treatment options, proton beam doses and principles. 

Author Response

I congratulate authors for sharing their experience with clival chordomas and preparing the systematic review.

The retrospective review is adequate including surgical treatments before and after endonasal endoscopic era. Interestingly, endoscopic endonasal approaches were not correlated with increased survival or as a favorable prognostic factor.

The molecular biology segment is very welcomed and providing additional value to the retrospective review.

Response: We thank the Reviewer for having appreciated our work.

I recommend expanding the discussion on the different radiation treatment options, proton beam doses and principles.

Response: We expanded the discussion on radiation treatment options (Discussion section, Page 18, lines 316-328).

Round 2

Reviewer 1 Report

Thank you for the edits. No further comments.

Improved. The editors may have some suggestions